# Short- versus long-term dual antiplatelet therapy after second-generation drug-eluting stent implantation in patients with diabetes mellitus: A meta-analysis of randomized controlled trials

Hongyu Zhang[◎], Junsong Ke[◎], Jun Huang[iD]*, Kai Xu, Yun Chen

Department of Cardiology, The First Affiliated Hospital of Nanchang University, Nanchang, Jiangxi, China

◎ These authors contributed equally to this work.
* junhuang918@163.com

**Data Availability Statement:** Data from PubMed, Cochrane Library, EMBASE and ClinicalTrials.gov.

## Abstract

### Background

Diabetes is considered to be a high-risk factor for thromboembolic events. However, available data about the optimal dual antiplatelet therapy (DAPT) in patients with diabetes mellitus (DM) after second-generation drug-eluting stent (DES) implantation are scant.

### Objective

The purpose of this study was to compare the impact of various DAPT durations on clinical outcomes in patients with DM after second-generation DES implantation.

### Methods

We searched PubMed, Embase, and the Cochrane Library for studies that compared short-term ($\leq$ 6 months) and long-term ($\geq$ 12 months) DAPT in patients with DM. The primary endpoints were late (31–365 days) and very late (> 365 days) stent thrombosis (ST). The secondary endpoints included myocardial infarction (MI), target vessel recanalization (TVR), all-cause death, and major bleeding.

### Results

Six randomized controlled trials, with a total of 3,657 patients with DM, were included in the study. In terms of the primary endpoint, there was no significant difference between the two groups in late (OR 1.15, 95% CI: 0.42–3.19, $P = 0.79$) or very late (OR 2.18, 95% CI: 0.20–24.18; $P = 0.53$) ST. Moreover, there was no significant difference in the secondary endpoints, including MI (OR 1.11, 95% CI: 0.72–1.71, $P = 0.63$), TVR (OR 1.31, 95% CI: 0.82–2.07, $P = 0.26$), all-cause death (OR 1.03, 95% CI: 0.61–1.75, $P = 0.90$) and major bleeding (OR 1.07, 95% CI: 0.34–3.40, $P = 0.90$) between the two groups.

More information about the data sources used is in the Supporting Information files.

**Funding:** The authors received no specific funding for this work.

**Competing interests:** The authors have declared that no competing interests exist.

## Conclusion

Our study demonstrated that compared with long-term DAPT, short-term DAPT had no significant difference in the clinical outcomes of patients with DM implanted with second-generation DES.

## Introduction

Nowadays, drug-eluting stent (DES) is widely used in diabetic patients with coronary artery disease because of their lower restenosis and target lesion revascularization rates compared to bare-metal stents (BMS). However, due to incomplete endothelial coverage, DES appear to be associated with an increased risk of late and very late stent thrombosis (ST) [1]. Compared with the early generation DES, the second-generation DES significantly reduces ST as these stents are made of novel materials, and newer anti-proliferative drugs are used. Unfortunately, the risk of thrombosis after second-generation DES implantation in patient with diabetes mellitus (DM) remains high.

As is well known, DM is an independent risk factor for ST after a percutaneous coronary intervention (PCI) [2]. Due to the high platelet reactivity, increased thrombin activity, and decreased reactivity of antiplatelet drugs in patients with DM, these factors might lead to ischemic events [3]. Therefore, how to prevent thromboembolism and adverse cardiovascular events after PCI in patients with DM is still worth investigating.

Current guidelines recommend dual antiplatelet therapy (DAPT) of aspirin and P2Y12 inhibitor after stent implantation [4, 5]. This dual therapy aims to reduce the risk of ST after PCI and prevent the occurrence of thromboembolic events in coronary arteries outside the stented segment. However, available data on the optimal DAPT in patients with DM after second-generation DES implantation are scant. Moreover, with high diabetes prevalence worldwide, the number of patients with DM, complicated by coronary artery disease is increasing, introducing a great challenge to clinical practice. Therefore, we conducted this meta-analysis to investigate the optimal DAPT in patients with DM after PCI.

## Methods

### Data sources and search strategy

We searched PubMed, Embase, and the Cochrane Library for studies that compared short-term ($\leq$ 6 months) and long-term ($\geq$12 months) DAPT in patients with DM after second-generation DES implantation. We also searched for relevant trials in the ClinicalTrials.gov. The search strategy is shown in S1 Text. Search terms included "diabetes mellitus", "diabetes," "percutaneous coronary intervention", "drug-eluting stent", "dual antiplatelet therapy", "aspirin", "clopidogrel", "prasugrel", "ticagrelor", and "P2Y12 receptor inhibitor". We searched all articles in the databases up to December 2019. Only studies published in English were included in this study. Ethical approval and patient consent were not required because this is an analysis of previously published studies.

### Inclusion criteria

randomized controlled trials (RCTs); (2) trials that compared short-term ($\leq$ 6 months) and long-term ($\geq$ 12 months) DAPT following PCI; (3) trials on implanted second-generation DES; (4) trials that report clinical outcomes in patients with DM, such as ST, myocardial infarction (MI), target vessel recanalization (TVR), all-cause death, or major bleeding.

### Exclusion criteria

Meta-analysis, case reports, ongoing trials or editorials; (2) studies that did not compare short-term($\leq$ 6 months) and long-term($\geq$ 12 months) DAPT after PCI; (3) studies that did not include patients with DM; (4) studies that did not include second-generation DES; (5) studies that did not report adverse clinical endpoints as described in the inclusion criteria; (6) duplicate studies.

### Study endpoints

The primary endpoints were late and very late ST. Secondary endpoints included MI, TVR, all-cause death, and major bleeding. All clinical endpoint definitions followed the original definitions in included the studies. ST was as defined by the Academic Research Consortium [6]. Late ST was defined as ST that occurred between 31 to 365 days after PCI. Very late ST was defined as ST that occurred more than 365 days after PCI. Major bleeding was defined differently in each study. In the RESET trial [10], it was defined following the Thrombolysis in Myocardial Infarction (TIMI) criteria [7]. In the OPTIMIZE trial [11], it referred to intracranial, intraocular, or retroperitoneal hemorrhages. In the SECURITY trial [19], it was classified according to the standardized definition of the Bleeding Academic Research Consortium (BARC) [8]. We accepted the definition of major bleeding as used in each study.

### Data extraction and risk of bias assessment

Two investigators (ZHY and KJS) independently reviewed the titles and abstracts to excluded irrelevant records, and then obtained eligible articles. Disagreements were resolved by consensus, and a third opinion (XK) was sought if necessary. After agreement on the included studies, relevant data were extracted from the studies, including study characteristics, baseline patient characteristics, and clinical outcomes. Quality assessment of the above studies was based on the Cochrane Collaboration's risk of bias tool [9].

### Statistical analyses

All data analyses were conducted by the Cochrane Review Manager (RevMan 5.3). Results are expressed as odds ratios (OR) with its 95% confidence intervals (CI). Heterogeneity was evaluated by Cochran's Q test, and $P$-value $< 0.05$ was considered statistically significant. The $I^2$ statistic test was also used to assessed heterogeneity, in which $I^2 < 25\%$, $25\% \leq I^2 \leq 50\%$, $I^2 \geq 50\%$ were considered low, medium, or high heterogeneity, respectively. If $I^2$ was more than 50%, a random-effects model was used for data analysis. Otherwise, a fixed model was used.

## Results

### Search result and study characteristics

According to the search strategy, a total of 5,193 records were identified. After removing 672 duplicates, we carefully reviewed the titles and abstracts and eliminated 4,490 irrelevant studies. Thirty-one full-text articles were assessed for eligibility, and six RCTs [10–15] were finally included in our study. The study selection process is shown in Fig 1.

A total of 3,657 patients with DM were involved, including 1,813 patients in the short-term DAPT group and 1,844 patients in the long-term DAPT group. All patients were implanted with a second-generation DES, such as everolimus-eluting stent (EES), biolimus-eluting stents (BES), or zotarolimus-eluting stents (ZES). The study published by Tarantini et al. [14] was a sub-study of the SECURITY trial [19], and all participants were diabetic patients. The remaining five articles were subgroup analyses of RCTs [10–13, 15]. Two studies compared 3-month

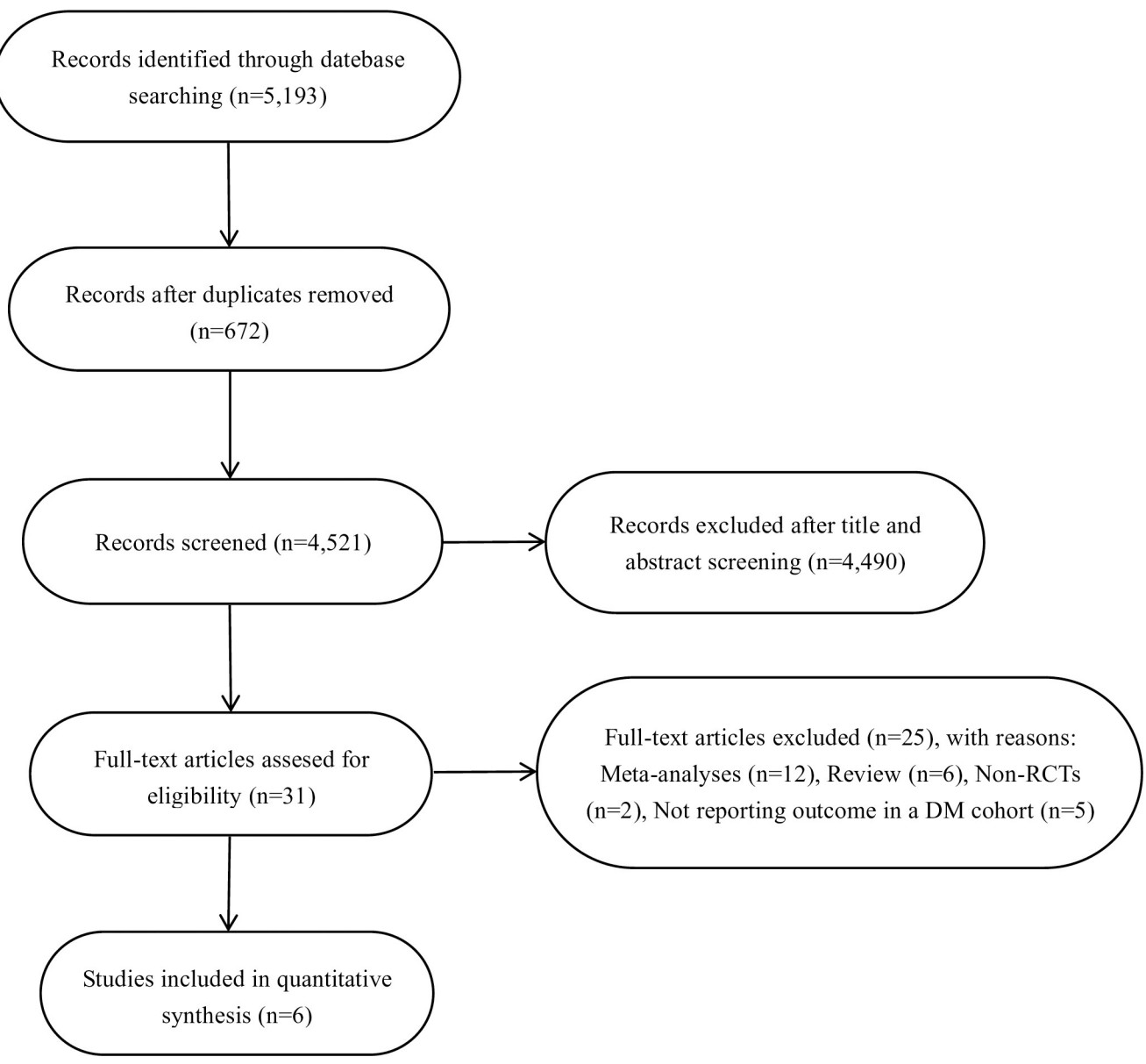

**Fig 1. Flowchart for study selection.**

*versus* 12-month DAPT [10, 11], one study compared 6-month *versus* 24-month DAPT [12], and three studies compared 6-month *versus* 12-month DAPT [13–15]. Most studies used clopidogrel as a second antiplatelet agent, with follow-up periods ranging from 12 to 24 months. The general characteristics of all studies are listed in Table 1.

## Patient characteristics of included studies

Table 2 describes the patient characteristics of included studies. In this meta-analysis, the average age of patients ranged from 60.0 to 66.7 years; the percentage of men between 62.9% and 80.8%; diabetic patients between 22.1% and 100%; and the percentage of patients with acute coronary syndrome (ACS) between 31.6% and 100%. According to the general characteristics of patients, there was no difference between the short- and long-term DAPT groups.

Table 1. Characteristics of included trials.

| Study name | Year | Type of study | Types of participants | No. of DM patients in the short-term group (n) | No. of DM patients in the long-term group (n) | Type of DES used | Comparison of DAPT duration (months) | Clopidogrel | Ticagrel or prasugrel | Follow up period |
|---|---|---|---|---|---|---|---|---|---|---|
| RESET [10] | 2012 | RCT | ACS and SCAD | 146 | 146 | ZES | 3 vs 12 | 100% | 0% | 12 months |
| OPTIMIZE [11] | 2013 | RCT | ACS and SCAD | 554 | 549 | ZES | 3 vs 12 | 100% | 0% | 12 months |
| ITALIC [12] | 2015 | RCT | ACS and SCAD | 331 | 344 | EES | 6 vs 24 | 98.6% | 1.4% | 12 months |
| I-LOVE IT 2 [13] | 2016 | RCT | ACS and SCAD | 211 | 203 | BP-DES | 6 vs 12 | 100% | 0% | 18 months |
| Tarantini 2016 [14] | 2016 | RCT | ACS and SCAD | 206 | 223 | ZES, EES, BES | 6 vs 12 | 98.8% | 1.2% | 24 months |
| SMART-DATE [15] | 2018 | RCT | ACS | 365 | 379 | ZES, EES, BES | 6 vs 12 | 80.8% | 19.2% | 18months |

ACS: acute coronary syndrome; BES: biolimus-eluting stents; BP-DES: biodegradable polymer drug eluting stent; DAPT: dual anti-platelet therapy; DM: diabetes mellitus; DES: drug-eluting stents; EES: everolimus-eluting stent; ZES: zotarolimus-eluting stents; RCT: randomized controlled trials; SCAD: stable coronary artery disease; ZES: zotarolimus-eluting stents.

### Risk of bias assessment and sensitivity analysis

The quality of all studies was assessed according to the Cochrane Collaborative's tool, as shown in S1 Table. Risk of bias summary and graph (Fig 2) showed that all included studies in this meta-analysis were in the lower categories for risk of bias. Publication bias was assessed by funnel plots. Evidence of publication bias reported among the studies that assessed all clinical endpoints was also low as shown in S1–S5 Figs. Sensitivity analysis was performed by removing one study at a time and repeating the statistical analysis. The statistical significance of the overall results did not change through the sensitivity analyses, confirming their robustness.

### Clinical endpoints

The primary endpoints of this study were late and very late ST after PCI. For late ST, there were six patients (0.4%) in the short-term DAPT group and five patients (0.3%) in the long-term DAPT group. Only two studies reported very late ST incidence, which occurred in two patients (0.5%) in the short-term DAPT group and one patient (0.2%) in the long-term DAPT group. There was no significant difference in late (OR 1.15, 95% CI: 0.42–3.19, $P = 0.79$, $I^2 = 0\%$, Fig 3A) or very late (OR 2.18, 95% CI: 0.20–24.18; $P = 0.53$, Fig 3B) ST between the two groups.

In terms of MI, there were 44 patients (2.4%) in the short-term DAPT group and 40 patients (2.2%) in the long-term DAPT group. For TVR, there were 33 patients (2.3%) in the long-term DAPT group and 43 patients (3.0%) in the short-term DAPT group. For all-cause death, there were 29 patients (2.0%) in the short-term DAPT group, and 28 patients (1.9%) in the long-term DAPT group. For Major bleeding, there were four patients (0.5%) in the short-term DAPT group and six (0.6%) in the long-term DAPT group. There were no significant differences between the two groups in the rates of MI (OR 1.11, 95% CI: 0.72–1.71; $P = 0.63$, $I^2 = 0\%$, Fig 3C), TVR (OR 1.31, 95% CI: 0.82–2.07; $P = 0.26$, $I^2 = 0\%$, Fig 3D), all-cause death (OR 1.03, 95% CI: 0.61–1.75; $P = 0.90$, $I^2 = 0\%$, Fig 3E), or major bleeding (OR 1.07, 95% CI: 0.34–3.40; $P = 0.90$, $I^2 = 0\%$, Fig 3F).

**Table 2. Patient characteristics of included studies.**

| Characteristics | RESET [10]* | | OPTIMIZE [11]* | | ITALIC [12]* | | I-LOVE IT 2 [13]* | | Tarantini 2016 [14] | | SMART-DATE [15]* | |
|---|---|---|---|---|---|---|---|---|---|---|---|---|
| | ST | LT | ST | LT | ST | LT | ST | LT | ST | LT | ST | LT |
| Age (years) | 62.4±9.4 | 62.4±9.8 | 61.3±10.4 | 61.9±10.6 | 61.7±10.9 | 61.5±11.1 | 60.4±10.2 | 60.0±10.0 | 65.5±10.1 | 66.7±9.1 | 62.0±11.5 | 62.2±11.9 |
| Males (%) | 64.9 | 62.9 | 63.5 | 63.1 | 80.8 | 79.2 | 67.2 | 68.7 | 71.8 | 74.0 | 74.9 | 75.9 |
| Hypertension (%) | 62.6 | 61.4 | 86.4 | 88.2 | 65.2 | 64.7 | 61.0 | 64.8 | 82.5 | 80.3 | 49.9 | 48.7 |
| Dyslipidemia (%) | 58.2 | 59.9 | 63.2 | 63.7 | 67.1 | 67.1 | 25.3 | 23.4 | 69.4 | 70.9 | 24.2 | 25.2 |
| Diabetes (%) | 30.1 | 28.8 | 35.4 | 35.3 | 36.3 | 37.8 | 23.2 | 22.1 | 100 | 100 | 26.9 | 28.1 |
| Insulin dependent | - | - | 10.2 | 10.4 | - | - | 9.7 | 7.2 | 21.4 | 19.7 | - | - |
| Current smoking (%) | 25.2 | 22.8 | 18.3 | 17.3 | 50.9 | 52.7 | 36.6 | 38.3 | 33.5 | 35.9 | 38.0 | 40.1 |
| Previous MI (%) | 1.8 | 1.6 | 34.6 | 34.8 | 15.6 | 14.7 | 17.2 | 15.8 | 23.8 | 17.1 | 2.3 | 1.7 |
| Previous PCI (%) | 3.7 | 3.0 | 20.9 | 19.1 | 24.1 | 22.5 | 8.5 | 6.5 | 22.8 | 17.0 | 4.9 | 3.9 |
| Previous CABG (%) | 0.2 | 0.6 | 7.1 | 8.2 | 6.7 | 4.9 | 0.4 | 0.4 | 5.8 | 7.2 | - | - |
| LVEF (%) | 64.2±9.4 | 63.9±9.4 | - | - | - | - | 60.8±8.4 | 60.3±8.2 | 55.8±9.7 | 55.7±9.1 | 55.5±11.0 | 55.4±10.5 |
| Clinical presentation (%) | | | | | | | | | | | | |
| STEMI | 14.7 | 13.8 | - | - | 7.3 | 7.6 | 13.4 | 13.7 | 0.0 | 0.0 | 37.5 | 37.9 |
| NSTEMI | 0.0 | 0.0 | 5.4 | 5.4 | 15.7 | 16.5 | 11.3 | 10.7 | 0.0 | 0.0 | 31.5 | 31.4 |
| Unstable angina | 40.8 | 39.9 | - | - | 20.3 | 20.1 | 58.0 | 56.5 | 35.9 | 32.3 | 31.0 | 30.7 |
| Stable angina | 44.5 | 46.3 | 59.8 | 58.6 | 41.3 | 41.5 | 14.3 | 15.1 | 64.1 | 67.7 | 0.0 | 0.0 |
| Silent ischemia | 0.0 | 0.0 | 8.6 | 9.2 | 15.4 | 14.3 | 3.0 | 4.0 | 0.0 | 0.0 | 0.0 | 0.0 |
| Number of lesions (%) | | | | | | | | | | | | |
| 1-vessel disease | 56.9 | 57.1 | - | - | 50.3 | 54.3 | 68.4 | 69.0 | 45.6 | 52.0 | - | - |
| 2-vessel disease | 27.6 | 27.6 | - | - | 30.2 | 27.7 | 27.8 | 27.3 | 42.2 | 34.5 | - | - |
| 3-vessel disease | 15.5 | 15.3 | - | - | 19.5 | 18.0 | 3.4 | 3.5 | 12.1 | 13.5 | - | - |
| Treated vessel (%) | | | | | | | | | | | | |
| Left anterior descending | 52.7 | 53.6 | 47.9 | 46.6 | 73.4 | 72.3 | 45.9 | 45.3 | 39.0 | 41.0 | 56.6 | 61.0 |
| Left Circumflex | 21.3 | 19.2 | 23.4 | 24.3 | 50.0 | 47.9 | 22.9 | 22.2 | 39.0 | 41.0 | 24.4 | 25.1 |
| Right coronary artery | 67.6 | 69.2 | 27.6 | 27.7 | 53.6 | 52.1 | 29.4 | 30.8 | 19.0 | 18.0 | 37.2 | 36.2 |
| Bifurcation (%) | 0.0 | 0.0 | 14.7 | 14.9 | - | - | 30.8 | 33.1 | 10.7 | 12.1 | 9.2 | 9.1 |
| Stents implanted, mean | | | | | | | | | | | | |
| Per patient | 1.3 | 1.5 | 1.6 | 1.6 | 1.7 | 1.7 | 1.7 | 1.7 | 1.6 | 1.6 | 1.4 | 1.5 |
| Per lesion | 1.0 | 1.2 | 1.2 | 1.2 | - | - | 1.3 | 1.3 | 1.1 | 1.2 | 1.1 | 1.1 |
| Stent length per lesion (mm) | 22.7 | 22.9 | 20.4 | 20.4 | - | - | 30.2 | 30.5 | 19.2 | 19.3 | 26.1 | 26.3 |

*The characteristics of the patients were extracted from the original text, include diabetic and non-diabetic patients; CABG: Coronary artery bypass grafting; LVEF: Left ventricular ejection fraction; LT: Long term DAPT group; MI: Myocardial infarction; PCI: Percutaneous coronary intervention; ST: Short term DAPT group.

## Discussion

Our study aimed to evaluate the safety and efficacy of short-term ($\leq$ 6 months) *versus* long-term ($\geq$ 12 months) DAPT in patients with DM after second-generation DES implantation. The results show that the different DAPT durations had no impact on the clinical outcomes of patients with DM in terms of late and very late ST, MI, TVR, all-cause death, and major bleeding.

Unlike BMS, DES is widely used in clinical practice because it can inhibit the proliferation of smooth muscle cells and reduce the rate of restenosis. However, due to insufficient endothelialization, DES appears to be associated with an increased propensity towards ST [1]. This tendency is especially true for the early generation DES. Pfisterer et al. [16] reported that the risk for very late ST in early generation DES was twice that of BMS (2.6% vs. 1.3%) and was even

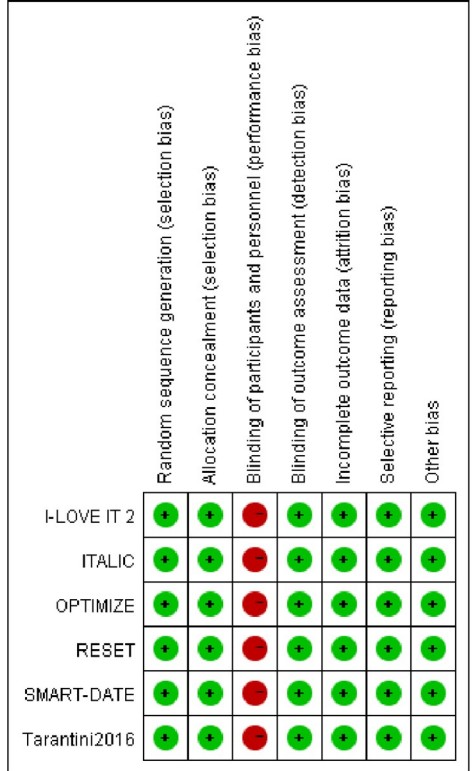

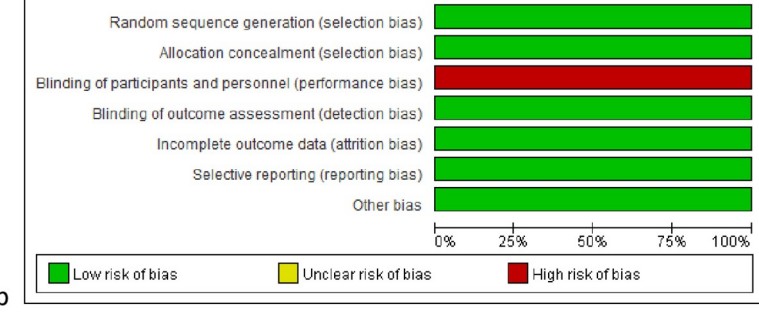

**Fig 2. Risk of bias graph and summary.** (a) Risk of bias graph: review of the authors' judgments about each risk of bias item, presented as percentages across all of the included studies. (b) Risk of bias summary: review of the authors' judgments about each risk of bias item for each included study.

higher in cases of early discontinuation of DAPT (4.9%). Such data led to the view that patients with DES should take antiplatelet drugs as long as possible. However, with the advent of the second-generation DES, some trials had demonstrated the safety of short-term DAPT after PCI [10–15, 17–19]. Because the second generation DES was implemented with new materials and new anti proliferation drugs, such as zoltamox, everolimus and bioramus. Compared with early DES, these factors greatly reduce the risk of ST. From this point of view, it seems reasonable to shorten the duration of DAPT.

Patients with ACS should consider receiving DAPT for at least 12 months after PCI, a currently recognized clinical practice [4, 5]. However, recent evidence indicates that the benefits of thrombosis prevention endowed by DAPT following DES implantation are primarily in the first six months after implantation [20]. Some special patient groups might, however, benefit

## a) Late tent thrombosis

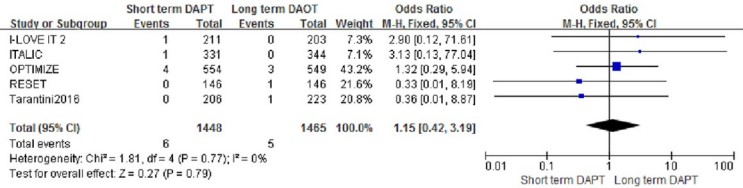

## b) Very late stent thrombosis

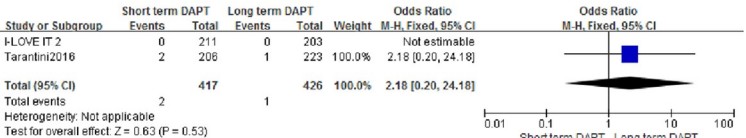

## c) Myocardial infarction

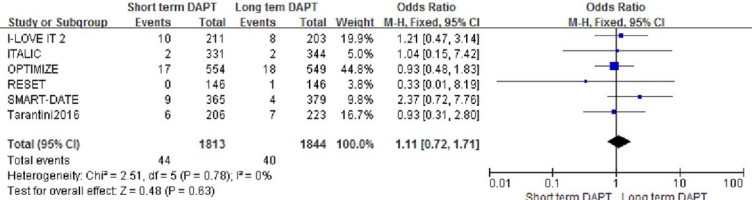

## d) Target vessel recanalization

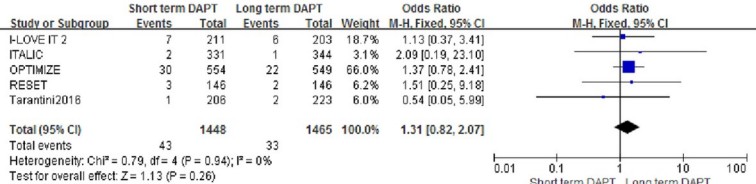

## e) All-cause death

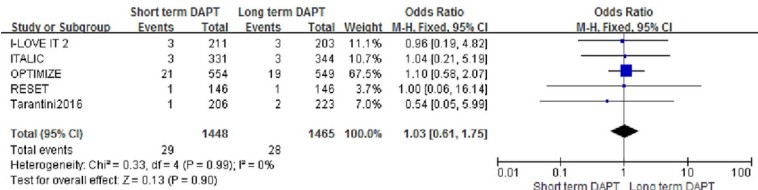

## f) Major bleeding

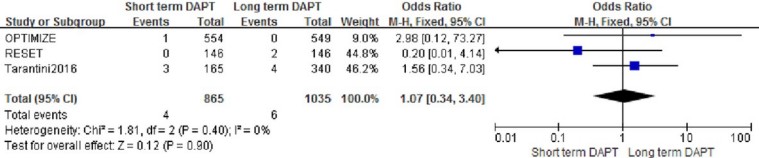

**Fig 3. Forest plot of late and very late stent thrombosis, myocardial infarction, target vessel revascularization, all-cause death and major bleeding.**

from long-term DAPT. It is well known that DM is consider to be a prothrombotic state characterized by platelet activation, inflammation, and hypercoagulability [3]. Platelet activation is a key factor in thrombosis in patients with DM, and even in the early and preclinical stages of patients with impairment of glucose metabolism, platelet activation increases with increased thromboxane biosynthesis [3, 21]. These findings led to the view that prolonging the DAPT is a more reasonable treatment approach for DM patients after PCI.

In this context, some studies have investigated the optimal duration of DAPT after stent implantation. In the EXCELLENT trial, when compared with six months of DAPT, twelve months of DAPT will significantly reduce the target vessel failure (composed of cardiac death, MI, and TVR) rate of patients with DM [22]. Unfortunately, 25% of the patients in this trial were implanted with early generation DES, which might have affected the robustness of the results. On the other hand, trials in which second-generation DES was used, such as OPTI-MIZE [11], ITALIC [12], I-LOVE-IT 2 [13] and SECURITY [19], showed that long-term DAPT did not improve adverse clinical outcomes in patients with DM, including all-cause death, MI, ST, and TVR. Partially similar to the results of these trials, a recent meta-analysis involving 17 RCTs also showed that in patients with ACS and stable coronary heart disease, DAPT for three to six months was safer than DAPT for twelve months. Notably, this meta-analysis involved more than 30% of the patients with DM. Not only that, their subgroup analysis found that, when compared with three to six months, long-term DAPT was associated with a higher all-cause mortality in patients implanted with second-generation DES [23]. However, in our analysis, we did not find a significant difference in all-cause mortality between the two groups. This might be because our analysis evaluated different trials and subgroups. Another meta-analysis published by Sharma et al. [24] found that the risks for MI, TVR, and ST were similar in patients with DM after PCI treated by short- or long-term DAPT. From these results, it is not difficult to see that long-term DAPT is not superior to short-term DAPT in preventing ischemic events in patients with DM after PCI.

For DAPT, it is essential to weigh the risk of ischemia and bleeding, as these are closely related to the occurrence of adverse events after PCI [25, 26]. When the duration of DAPT is prolonged, the subsequent concern is the risk of bleeding. More importantly, Berardis et al. [27] found that patients with DM had a higher risk of bleeding than those without DM, which was independent of the use of antiplatelet drug use. Recently, Bundhun et al. used meta-analysis to compare short- and long-term DAPT after DES implantation. Their analysis involved 15 studies and 25,742 diabetic patients. They showed that as the duration of DAPT was prolonged, bleeding, as defined by BARC, had increased significantly [28]. However, we found no difference in major bleeding events between the two groups in our meta-analysis. This might be due to the overall low incidence of overall bleeding events (0.5%). Despite this, other studies have also shown that long-term DAPT was associated with a higher risk of bleeding compared with short-term DAPT [28, 29]. A study published by Gargiulo et al. [29], concluded that diabetes should not be a driver for prolonging the DAPT, because the potential benefits of this strategy were accompanied by an increased risk of bleeding. Furthermore, Capodanno et al. [30] Proposed such a concept that in the contemporary era of widespread DES use, bleeding has a more significant impact on mortality than ST. From this perspective, it might be reasonable to shorten the duration of DAPT.

Our results corroborate with those reported by the International ISAR 2000 All Comers Registry, which showed that short- and long-term DAPT presented a similar risk for late ST,

MI, and all-cause death [31]. Based on these findings, they suggested that short-term DAPT was a reasonable strategy for patients with DM after PCI. All participants in their study were DM patients; however, it was not included in our meta-analysis as it was an observational trial.

## Limitations

There are several limitations to our study. First, only six RCTs were included in our meta-analysis, with a relatively small number of patients, different follow-up period, and different stent types, all of which could have affected the results. However, the included studies were strictly selected, so we could not avoid this limitation. Second, most of the six trials used aspirin combined with clopidogrel as DAPT, so we were unable to evaluate the effect of DAPT duration for other antiplatelet agents, such prasugrel or ticagrelor. Third, five of the include studies we included were subgroup analyses of large RCTs. These did not report the proportion of ACS in patients with DM. Therefore, our study could not individually assess the efficacy and safety of diabetic patients with ACS. Fourth, due to the low risk of thromboembolic events in the included patients, our results might not be apply to high-risk patients, such as those with DM who require insulin therapy, having lower extremity arterial disease, or experienced previous stent thrombosis.

## Conclusions

Although DM is considered a predictor of cardiovascular adverse events, our study demonstrated that long- and short-term DAPT had similar clinical outcomes in patients with DM after second-generation DES implantation. Based on our results, short-term DAPT seems to be a reasonable strategy for these patients. Because this short-term treatment reduces bleeding risk without increasing the occurrence of ST or other adverse clinical outcomes. However, due to the limitations of this study, our results need to be confirmed by more extensive RCTs for patients with DM.

## Supporting information

**S1 Fig. Funnel plot for late stent thrombosis.** OR: odds ratio; SE: standard error.
(TIF)

**S2 Fig. Funnel plot for myocardial infarction.** OR: odds ratio; SE: standard error.
(TIF)

**S3 Fig. Funnel plot for target vessel revascularization.** OR: odds ratio; SE: standard error.
(TIF)

**S4 Fig. Funnel plot for all-cause death.** OR: odds ratio; SE: standard error.
(TIF)

**S5 Fig. Funnel plot for major bleeding.** OR: odds ratio; SE: standard error.
(TIF)

**S1 Table. Risk of bias assessment in details.**
(DOCX)

**S2 Table. Checklist for PRISMA guidelines.**
(DOC)

**S1 Text. Search strategy.**
(DOCX)

## Acknowledgments

We would like to thank all the authors in particular.

## Author Contributions

**Conceptualization:** Hongyu Zhang, Junsong Ke, Jun Huang.

**Data curation:** Hongyu Zhang, Junsong Ke, Kai Xu.

**Formal analysis:** Hongyu Zhang, Junsong Ke, Kai Xu, Yun Chen.

**Investigation:** Hongyu Zhang, Junsong Ke, Kai Xu, Yun Chen.

**Methodology:** Hongyu Zhang.

**Supervision:** Junsong Ke, Kai Xu.

**Visualization:** Yun Chen.

**Writing – original draft:** Hongyu Zhang, Junsong Ke.

**Writing – review & editing:** Hongyu Zhang, Junsong Ke.

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
