## [Decision Letter · Decision Letter 0]

2 Sep 2020

PONE-D-20-24066

Short- versus long-term  dual antiplatelet therapy after second-generation drug-eluting stent implantation in patients with diabetes mellitus: A Meta-analysis of Randomized Controlled Trials

PLOS ONE

Dear Dr. Huang,

Thank you for submitting your manuscript to PLOS ONE. After careful consideration, we feel that it has merit but does not fully meet PLOS ONE’s publication criteria as it currently stands. Therefore, we invite you to submit a revised version of the manuscript that addresses the points raised during the review process.

Editors comments:

The major limitation of this study is that it was snot mentioned whether this is all PCI with diabetes  and percentage of ACS or stable ischemic heart disease ( SIHD) patients. The current guidelines have specific recommendations on DAPT such as ACS patients 6-12 month and stable angina patients 3-6 months and can stop after minimum 3 or 6 months if needed. 

I recommend to do 1) a subgroup analysis of ACS and SIHD patients 2) 3 months Vs 6 months instead of 6 months vs 12 months 

Please submit your revised manuscript by 60 days.  If you will need more time than this to complete your revisions, please reply to this message or contact the journal office at plosone@plos.org. Please include the following items when submitting your revised manuscript:

We look forward to receiving your revised manuscript.

Kind regards,

Timir Paul

Academic Editor

PLOS ONE

Journal Requirements:

2. Please provide a quality a publication bias assessment on the studies included in the systematic review

4.

We note that you have indicated that data from this study are available upon request. PLOS only allows data to be available upon request if there are legal or ethical restrictions on sharing data publicly. For information on unacceptable data access restrictions, please see http://journals.plos.org/plosone/s/data-availability#loc-unacceptable-data-access-restrictions.

5. Please amend the manuscript submission data (via Edit Submission) to include author Kai Xu. M.M., Yun Chen. M.M

6.

Please include your tables as part of your main manuscript and remove the individual files. Please note that supplementary tables (should remain/ be uploaded) as separate "supporting information" files

**Editor Comments :**

The major limitation of this study is that it was snot mentioned whether this is all PCI with diabetes  and percentage of ACS or stable ischemic heart disease ( SIHD) patients. The current guidelines have specific recommendations on DAPT such as ACS patients 6-12 month and stable angina patients 3-6 months and can stop after minimum 3 or 6 months if needed. 

I recommend to do 1) a subgroup analysis of ACS and SIHD patients 2) 3 months Vs 6 months instead of 6 months vs 12 months

Reviewers' comments:

Reviewer's Responses to Questions

**Comments to the Author**

1. Is the manuscript technically sound, and do the data support the conclusions?

Reviewer #1: Yes

Reviewer #2: Yes

2. Has the statistical analysis been performed appropriately and rigorously? 

Reviewer #1: Yes

Reviewer #2: I Don't Know

3. Have the authors made all data underlying the findings in their manuscript fully available?

Reviewer #1: Yes

Reviewer #2: Yes

4. Is the manuscript presented in an intelligible fashion and written in standard English?

Reviewer #1: Yes

Reviewer #2: Yes

5. Review Comments to the Author

Reviewer #1: This meta-analysis is well performed. I have some concern regarding the study as mentioned below.

- 6 months DAPT after PCI has already been accepted in the guidelines to be safe for all patients including diabetes. The question is not to compare 6 months of DAPT to 12 months or more, but rather to compared shorter DAPT therapy duration to 6 months DAPT therapy duration.

- The manuscript is lacking the brief overview of each research studies used in meta-analysis in tabular format including stent types, IDDM vs NIDDM if data is available, bifurcation stenting, anticoagulation used during PCI, duration of DAPT, follow-up time, etc. This is very vital for any meta-analysis. I see some details in the result section, but that is not enough.

Reviewer #2: besides few words corrections, it is a well written nice manuscript. It is written in good intellectual language and lays the strengths and weaknesses of the study. The limitations were laid out nicely and it speaks for the study power.

6. PLOS authors have the option to publish the peer review history of their article (what does this mean?). If published, this will include your full peer review and any attached files.

Reviewer #1: No

Reviewer #2: No

---

## [Author Response · Author response to Decision Letter 0]

8 Nov 2020

Reply to editor comments:

1)a subgroup analysis of ACS and SIHD patients 

 Response: Since the study we included did not report the proportion of diabetic patients with ACS or SIHC, we were unable to conduct a subgroup analysis of such patients. This point has been described in the third point of the limitations of this article

2)3 months vs 6 months instead of 6 months vs 12 months 

Response: Current clinical practice guidelines recommend 6–12-month DAPT following DES implantation. Despite these recommendations, due to refinements in DES technologies and the advent of potent P2Y12 receptor inhibitors, the optimal DAPT after stent implantation in patients with diabetes continues to be a matter of debate. Unfortunately, as literature is scant concerning DAPT durations for DM, the current meta-analysis was hindered in determining the impact of second-generation DESs or new-generation P2Y12 inhibitors on the optimal DAPT duration. We searched a large amount of literatures and found that most trials compared the DAPT of 3-6 months to 12 months or longer. Due to the limited literature available, we could not compare 3 months to 6 months of DAPT. And our meta-analysis shows that DAPT of 3-6 months is not inferior to DAPT of 12-24 months, which also has a certain guiding significance for clinical practice.

Reply to reviewer #1

1.6 months DAPT after PCI has already been accepted in the guidelines to be safe for all patients including diabetes. The question is not to compare 6 months of DAPT to 12 months or more, but rather to compared shorter DAPT therapy duration to 6 months DAPT therapy duration.

Response: At present, there are few literature about the duration of DAPT after PCI in patients with diabetes, and most trials compared the DAPT of 3-6 months to 12 months or longer, in order to explore the optimal DAPT duration for such patients. Due to limited data, we were unable to compare 6 months with shorter duration of DPAT. And our meta-analysis shows that for patients with stable coronary heart disease or ACS, 3-6 months of DAPT was noninferior to 12-24 months, which also has a certain guiding significance for clinical practice.

2. The manuscript is lacking the brief overview of each research studies used in meta-analysis in tabular format including stent types, IDDM vs NIDDM if data is available, bifurcation stenting, anticoagulation used during PCI, duration of DAPT, follow-up time, etc. This is very vital for any meta-analysis. I see some details in the result section, but that is not enough.

Response: Some tables have been modified, as shown in Table 1 and Table 2.

Reply to reviewer #2

Besides few words corrections, it is a well written nice manuscript. It is written in good intellectual language and lays the strengths and weaknesses of the study. The limitations were laid out nicely and it speaks for the study power.

Response: None.

---

## [Editor Report · Decision Letter 1]

11 Nov 2020

Short- versus long-term  dual antiplatelet therapy after second-generation drug-eluting stent implantation in patients with diabetes mellitus: A Meta-analysis of Randomized Controlled Trials

PONE-D-20-24066R1

Dear Dr. Huang,

We’re pleased to inform you that your manuscript has been judged scientifically suitable for publication and will be formally accepted for publication once it meets all outstanding technical requirements.

Kind regards,

Timir Paul

Academic Editor

PLOS ONE

Additional Editor Comments (optional):

All queries have been addressed appropriately

---

## [Editor Report · Acceptance letter]

1 Dec 2020

PONE-D-20-24066R1 

Short- versus long-term dual antiplatelet therapy after second-generation drug-eluting stent implantation in patients with diabetes mellitus: A Meta-analysis of Randomized Controlled Trials 

Dear Dr. huang:

I'm pleased to inform you that your manuscript has been deemed suitable for publication in PLOS ONE. Congratulations! Your manuscript is now with our production department. 

Kind regards, 

on behalf of

Dr. Timir Paul 

Academic Editor

PLOS ONE